# High throughput in vivo functional validation of candidate congenital heart disease genes in *Drosophila*

Jun-yi Zhu[1†], Yulong Fu[1†], Margaret Nettleton[1], Adam Richman[1], Zhe Han[1,2]*

[1]Center for Cancer and Immunology Research, Children's National Medical Center, Washington, United States; [2]Department of Pediatrics, The George Washington University School of Medicine and Health Sciences, Washington, United States

**Abstract** Genomic sequencing has implicated large numbers of genes and de novo mutations as potential disease risk factors. A high throughput in vivo model system is needed to validate gene associations with pathology. We developed a *Drosophila*-based functional system to screen candidate disease genes identified from Congenital Heart Disease (CHD) patients. 134 genes were tested in the *Drosophila* heart using RNAi-based gene silencing. Quantitative analyses of multiple cardiac phenotypes demonstrated essential structural, functional, and developmental roles for more than 70 genes, including a subgroup encoding histone H3K4 modifying proteins. We also demonstrated the use of *Drosophila* to evaluate cardiac phenotypes resulting from specific, patient-derived alleles of candidate disease genes. We describe the first high throughput in vivo validation system to screen candidate disease genes identified from patients. This approach has the potential to facilitate development of precision medicine approaches for CHD and other diseases associated with genetic factors.

*For correspondence: zhan@childrensnational.org

†These authors contributed equally to this work

Competing interests: The authors declare that no competing interests exist.

## Introduction

A major problem confronting large-scale genomic sequencing studies of disease patients is to identify those genetic variants that actually represent disease-causing mutations. This is particularly true for candidate genes that have not previously been evaluated in an experimental model system for involvement in disease-relevant biological processes. Functional gene validation systems in model organisms are therefore essential. While a definitive assignment of disease association should be established through the analysis of mutant gene phenotypes in a mammalian model, this approach is not practical for rapid high-throughput initial screening of large numbers of candidate genes. *Drosophila*, by contrast, possesses the attributes of a simple animal model system that make it ideally suited for this role (*Bier, 2005*; *Chien et al., 2002*).

Use of *Drosophila* as a model to elucidate molecular disease mechanisms is well established and amply documented (*Cagan, 2011*; *Na et al., 2015*; *Zhang et al., 2013*; *Diop and Bodmer, 2015*; *Bier and Bodmer, 2004*; *Owusu-Ansah and Perrimon, 2014*). It has been established that 75% of human disease associated genes are represented in the fly genome by functional homologs (*Reiter et al., 2001*). Although it is difficult to directly link *Drosophila* developmental defects to patient symptoms, the *Drosophila* system can serve as a testing platform for gene function in development, which can be used to test disease association for a large number of candidate genes identified from patient genomic sequencing.

Congenital heart disease (CHD) affects 0.8% of live births (*Hoffman, 1990*, *2002*; *Reller et al., 2008*). Genetic factors are strongly implicated in CHD pathogenesis, but the great majority of genes (accounting for an estimated 75% of cases) remain unidentified (*Gelb et al., 2013*). *Drosophila* has

**eLife digest** Around one in 100 children are born with heart defects caused by congenital heart disease. Studying the genetic sequences of people with congenital heart disease has revealed many genes that may play a role in causing the condition, but few of these findings have been confirmed experimentally in animal model systems.

The fruit fly species *Drosophila melanogaster* is often used in genetic studies because it is a relatively simple organism. The insights gained from studying flies are often valuable for determining the direction of subsequent investigations in more complex animals – such as humans – that involve experiments that are more costly and less efficient.

Zhu, Fu et al. have now used fruit flies to investigate the effects of 134 genes that have been suggested to contribute to congenital heart disease. The investigation used a method that rapidly allowed the activity of specific genes to be altered in the flies. The effects that these alterations had on many aspects of heart development, structure and activity were then measured. Of all the genes tested, 70 caused heart defects in the flies. Several of these genes help to modify the structure of proteins called histones; these modifications play important roles in heart cell formation and growth.

Further tests showed that the effects of specific genetic errors that had been identified in people with congenital heart disease could be reliably reproduced in the flies. This may allow individual cases of congenital heart disease to be replicated and studied closely in the lab, helping to create treatments that are personalised to each patient.

Studying congenital heart disease in flies provides a fast and simple first step in understanding the roles that different genes play in the disease. Moving forward, precise gene editing techniques could be used to generate flies to examine the role of each of the genetic mutations that occur in individual patients. Ultimately, when gene editing techniques are ready to be used in humans, this could lead to cures for congenital heart disease at the DNA level, so that these mutations won't be passed on to the next generation.

served as a model to study genes related to CHD for over 20 years, based on the evolutionarily conserved genetic basis of heart development (*Bier and Bodmer, 2004*; *Olson, 2006*; *Yi et al., 2006*). The *Drosophila* heart (*Figure 1A*) is a rhythmically beating linear tube composed of parallel rows of fused contractile cardiac cells, flanked by adherent pericardial cells (*Vogler and Bodmer, 2015*). The heart functions to maintain circulation of hemolymph (insect 'blood') throughout the body cavity. In a typical beating cycle, hemolymph enters the posteriorly-located heart chamber in the abdomen and is pumped anteriorly through the aorta towards the head and brain. Although structurally of relatively low complexity, *Drosophila* and human heart development are both largely directed by the same highly conserved regulatory networks (*Vogler and Bodmer, 2015*). To demonstrate that the fly is the ideal in vivo animal model for cost-effective, rapidly informative, and efficient screening of candidate disease genes, we have developed a *Drosophila* validation system to screen candidate genes identified in a large-scale genomic sequencing study of congenital heart disease patients.

In 2013 the Pediatric Cardiac Genomics Consortium (PCGC) published a large-scale sequence analysis of sporadically occurring CHD cases (*Zaidi et al., 2013*), providing an opportunity to conduct systematic model system based experimental studies into the genetic basis of CHD, informed from the outset by clinical data. Such studies will ultimately facilitate the development of diagnostic, therapeutic, and preventive approaches, and precision medicine interventions (*Ashley, 2015*) for CHD. The initial report (*Zaidi et al., 2013*) of 249 protein-altering de novo mutations from 362 severe cases of CHD identified 223 candidate disease genes involved in diverse biochemical pathways, with 26 genes selected as particular genes-of-interest based upon bioinformatics criteria (*Zaidi et al., 2013*). We used *Drosophila* developmental genetics to quantitatively evaluate the phenotypes associated with heart-specific RNAi mediated gene silencing of fly homologs of mutated candidate disease genes. 174 genes (78%) had clear fly homologs, consistent with the published estimate of 75% conservation of disease gene homologs (*Reiter et al., 2001*). Interestingly, the 'top 26' genes identified by the PCGC study (*Zaidi et al., 2013*) all have conserved *Drosophila* homologs. Some of these genes were previously identified as having roles in *Drosophila* heart development, in

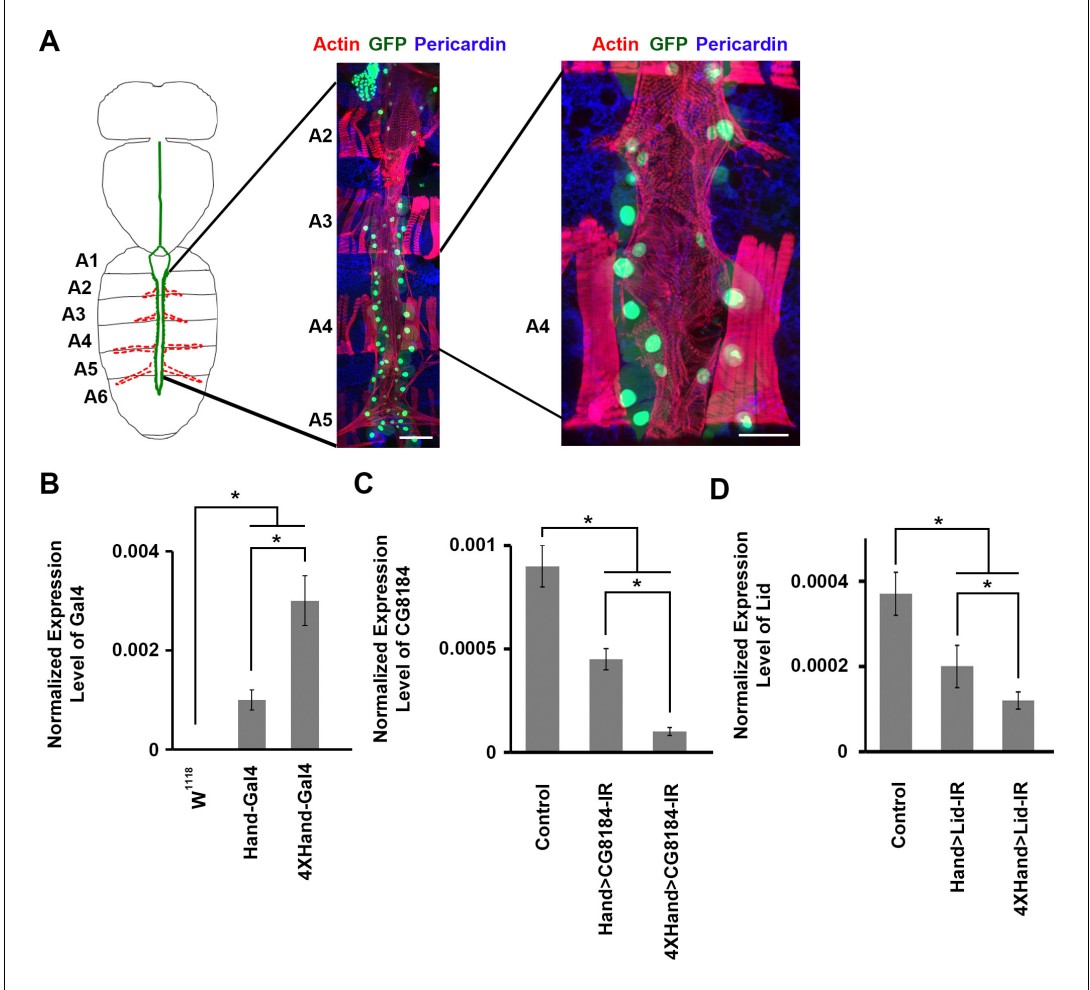

**Figure 1.** *Drosophila* adult heart structure and tissue visualization, evaluation of 1X vs. 4X *Hand* enhancer constructs driving heart specific silencing of gene expression. (**A**) *Drosophila* heart structure and visualization. **Left:** the adult heart is depicted schematically in green. Supporting lateral alary muscles are depicted in red. **Middle:** fluorescence microscopy of dissected heart tissue spanning abdominal segments A2–A5. Myocardial actin filaments (cardiac myofibers) were visualized by Phalloidin staining (red), which also stained somatic muscle fibers in segments A2, A3, and A4 and alary muscle fibers in A5. GFP (green) labels cardiomyocytes and pericardial nephrocytes. Expression of a nuclear-localized GFP transgene was controlled by the cardioblast specific *Hand* gene enhancer element. Scale bar = 100 μ. **Right:** higher magnification of segment A4 heart tissue. Scale bar = 50 μ. In this example, dissection and preparation for microscopy involved removal of a layer of longitudinal muscle, which resulted in the loss of some cardiomyocytes and pericardial nephrocytes. In situ, the heart tube is composed of parallel, symmetrical rows of cardiomyocytes (small nuclei), flanked by pericardial nephrocytes (large nuclei). (**B**) The 4XHand enhancer promotes significantly greater Gal4 mRNA production than the single Hand enhancer. The Gal4 mRNA level in the dissected adult heart was determined by qRT-PCR. Statistical significance (*) was defined as p<0.05. $w^{1118}$ was used as a negative control since it does not express any Gal4. (**C**) Compared to the single Hand enhancer, 4XHand induced significantly greater knockdown of CG8184 (the homolog of human *HUWE1*) mRNA when driving expression of the CG8184-IR RNAi silencing transgene. The CG8184 mRNA level in the dissected adult heart was determined by qRT-PCR. Statistical significance (*) was defined as p<0.05. (**D**) Compared to the single Hand enhancer, 4XHand induced significantly greater knockdown of Lid (the homolog of human *KDM5A* and *KDM5B*) mRNA when driving expression of the Lid-IR RNAi silencing transgene. The Lid mRNA level in the dissected adult heart was determined by qRT-PCR. Statistical significance (*) was defined as p<0.05. In **C** and **D**, Control flies were the progeny of a cross between homozygous 4XHand-Gal4 and $w^{1118}$, which has one copy of 4xHand-Gal4 but does not carry a UAS-RNAi silencing construct.

The following figure supplement is available for figure 1:

**Figure supplement 1.** 4XHand-Gal4 efficiently silences cardiac gene expression.

the context of a global in vivo RNAi screen for candidate fly heart genes. (*Neely et al., 2010*) The current study used a stronger heart-specific driver to validate candidate CHD genes.

We developed a highly efficient cardiac-targeted gene silencing approach in flies, and used this to examine effects on heart structure and function for fly homologs of 134 candidate disease genes published by the PCGC. We employed a quantitative phenotypic screening protocol that evaluated developmental lethality (pre-adult mortality), heart structure (*Figure 1A*; including morphology, cardiac myofibrillar density, cardiac collagen deposition, and cardioblast cell number) and adult longevity. We found that 52% of the tested fly homologs are required for cardiac development and function in flies. We also developed a gene replacement testing strategy involving simultaneous heart-specific silencing of an endogenous fly gene homolog and expression of either wild type or patient-derived mutant alleles of the candidate human disease gene.

## Results

To achieve a highly efficient heart-specific gene knock down, we generated a strong cardiac cell Gal4 driver featuring 4 tandem repeats of the *Drosophila Hand* gene cardiac enhancer (*Han and Olson, 2005*). We compared the levels of Gal4 RNA in heart tissues of transgenic flies carrying the new 4XHand-Gal4 driver construct versus flies carrying the original, established Hand-Gal4 driver (featuring a single *Hand* enhancer element). As anticipated, 4XHand promoted significantly higher heart cell expression of Gal4 mRNA (*Figure 1B*). We then compared the Hand-Gal4 and 4XHand-Gal4 drivers together with UAS-Gene-IR RNAi based silencing constructs (*Ni et al., 2008*, *2011*) for gene knock down efficiency when used to silence expression (i.e. lower heart cell RNA levels) of *Drosophila* genes CG8184 (*Figure 1C*) and *Lid* (*Figure 1D*). Consistent with the higher Gal4 expression, 4XHand was significantly more effective at silencing target gene expression through RNAi. To further confirm the utility of 4XHand-Gal4 for screening purposes, we comparatively evaluated the Hand-Gal4 and 4XHand-Gal4 constructs driving heart-specific expression of UAS-RNAi transgenes targeting eight *Drosophila* genes for developmental lethal (death at pre-adult stage) effects. The *Hand* enhancer is active during development from embryonic stages, and we reasoned that *Hand* enhancer-driven silencing of genes known to be essential for heart development would cause developmental lethality. For this assay, male and female flies of the appropriate genotypes were crossed to produce progeny carrying the indicated UAS-gene silencing (i.e. RNAi) interfering RNA (IR) construct, the expression of which was driven by Hand-Gal4 or 4XHand-Gal4. Progeny embryos were collected and allowed to develop under standard conditions. The results were quantitatively expressed as a Mortality Index (MI), the percentage of flies that died before adult emergence from the pupa stage. We found that for all eight tested genes, significant developmental lethal effects of heart-specific gene silencing were only observed in flies carrying a 4XHand-Gal4 driver (*Figure 1— figure supplement 1*).

We combined the new 4XHand-Gal4 driver with available fly lines carrying UAS-Gene-IR RNAi based silencing constructs to evaluate for heart specific essential function (i.e. MI) the homologs of all genes from the PCGC study for which multiple independent UAS-RNAi lines are available (134 total genes, *Supplementary file 1*). We extracted the data for the PCGC top 26 genes, identified as most-promising candidate disease genes on the basis of multiple bioinformatics-based criteria (*Figure 2A*). We reasoned that if these criteria alone were truly informative, the majority of these genes would yield higher MI scores, overall, than observed among all 134 tested genes representing the PCGC total list. As shown in *Figure 2* we observed MI values ranging across a spectrum spanning Normal ($\leq$6%, equivalent to nonbiased control genes), Low (7–30%), Medium (31–60%), to High (61–100%). Importantly, we found that the distribution of MI values across all ranges was not significantly different for the top 26 genes compared to the entire cohort of 134 candidate genes tested (*Figure 2A*). Thus, selection of candidate CHD genes on the basis of bioinformatics criteria alone did not enrich for genes inducing more severe developmental mortality phenotypes (indicative of more genes with essential roles in either heart development, tissue maintenance, or function). This observation emphasizes the importance of screening and selecting most-promising candidate genes on the basis of functional assays. To determine if the set of 134 candidate genes in itself can be demonstrated to be enriched for heart-essential genes, we examined previously obtained data from a screen of approximately 800 *Drosophila* genes silenced using the 4XHand-Gal4; UAS-RNAi system. These genes were randomly chosen to screen for effects on pericardial nephrocyte function

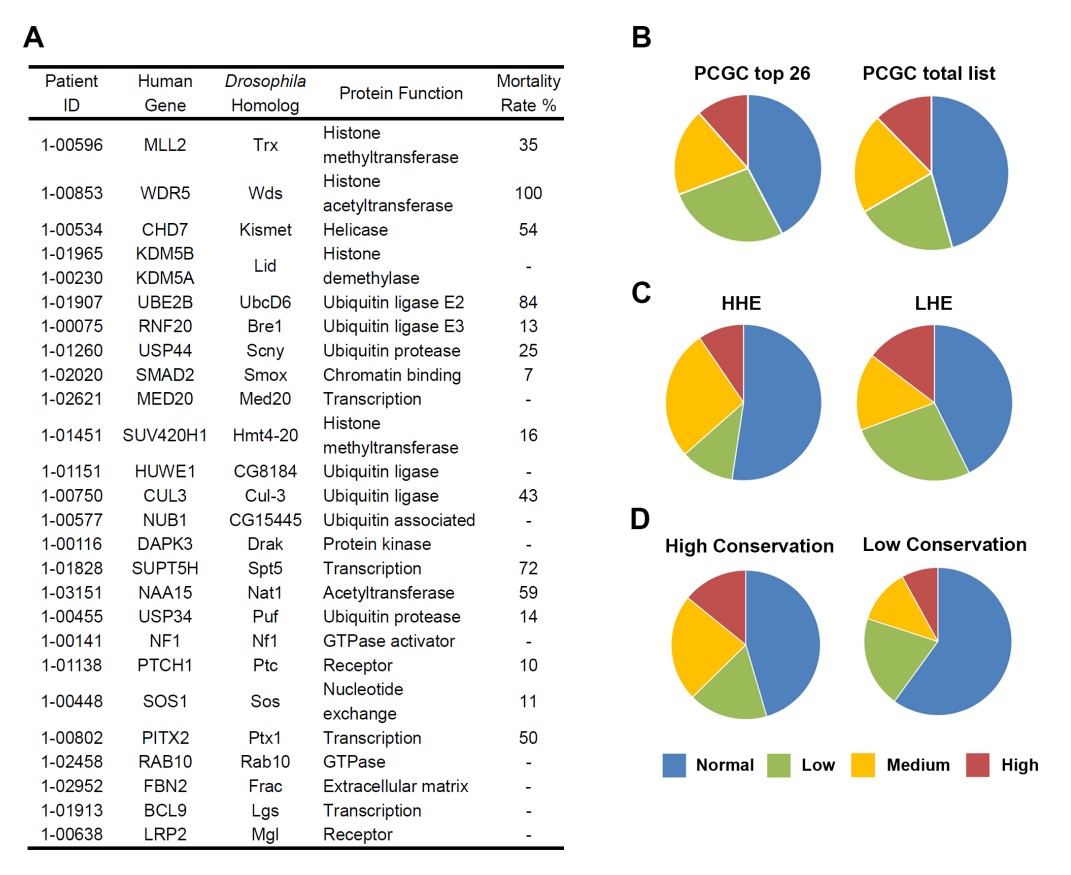

**Figure 2.** Top 26 candidate CHD genes and developmental lethality induced by heart specific RNAi-based silencing of *Drosophila* gene homologs. (A) 26 de novo mutated genes from CHD study participants selected as being of particular interest (*Zaidi et al., 2013*) based upon sequence quality, mutation type, the expression level of the mouse homolog during heart development on embryonic day 14.5, and previously reported involvement in CHD or heart development. The 26 corresponding *Drosophila* homologs are shown with protein function (Flybase), and Mortality Rate (Mortality Index). (B–D) The proportions of *Drosophila* gene homologs that, when silenced by cardiac cell specific RNAi expression, induce developmental lethality at normal/background levels (blue); low levels (green); medium levels (orange); high levels (red) based on Mortality Index values. The Mortality Index is determined by crossing homozygous UAS-RNAi transgenic flies with flies bearing a 4XHand-Gal4 'driver' (four repeats of the cardioblast cell-specific *Hand* enhancer element 5' of *Gal4*) balanced over *CyO*. Progeny flies that emerge as adults with curly wings (*CyO*, no transgene expression) vs. straight wings (expressing 4XHand-Gal4 driven UAS-RNAi transgene in cardioblasts) are recorded and the developmental mortality attributable to RNAi heart expression (Mortality Index) is calculated as (Curly – Straight) / Curly X 100. Divergence from 1:1 ratio (Normal, blue) $\geq$ 7% was considered a lethal phenotype. A Normal range of divergence from a 1:1 ratio of <6% based on analysis of 400 progeny from control crosses. Varying degrees of phenotype severity were observed (Low = 7–30%, green; Medium = 31–60%, orange; High = 61–100%, red). (B) Left: chart (PCGC top 26) shows proportions of RNAi silencing effects on lethality for *Drosophila* homologs of 26 genes identified as being of particular interest based exclusively on bioinformatics-based criteria. Right: chart (PCGC total list) shows the results of 134 fly homologs of all de novo mutated genes (with available RNAi silencing lines) identified in pediatric CHD study participants (*Zaidi et al., 2013*) (*Supplementary file 1*). (C) Comparison of silencing-induced lethality for *Drosophila* homologs of 134 candidate CHD-associated genes as a function of high (HHE) *versus* low (LHE) levels of expression of murine homologs in embryonic mouse heart. (D) Comparison of silencing-induced lethality for *Drosophila* homologs of 134 candidate CHD-associated genes as a function of fly-to-human gene conservation (High Conservation, score 6 to 10; Low Conservation, score 2 to 5).

(cardioblast lineage cells in which the *Hand* enhancer is active), so they are unbiased with respect to essential heart roles. We found that only 6% of these genes showed *any level* of developmental lethality when silenced (i.e. 94% normal). Thus, the PCGC genomic association study, though not gene selection based upon bioinformatics criteria, did enrich significantly for heart-essential genes as assayed in *Drosophila*. That such a small number of unbiased genes conferred a phenotype of silencing-induced developmental lethality also suggests that overexpression of 4xHand-Gal4 itself did not sensitize flies to RNAi mediated knockdown effects.

We also stratified and compared MI values we obtained for all of the 134 *Drosophila* genes as functions of expression level of murine homologs during mouse embryonic heart development, as reported in the PCGC study (and one basis for selection of top 26 genes) (*Zaidi et al., 2013*). As shown in *Figure 2C*, high heart expression (HHE) during development did not correlate with greater severity of developmental lethality associated with gene silencing in *Drosophila* heart. Contrary to expectation, low heart expression (LHE) of murine homologs was associated with relatively higher severity of *Drosophila* gene silencing. These observations again emphasize the importance of applying functional screening assays to the results of genomic sequencing studies. We noted nevertheless that the top 26 genes all have *Drosophila* homologs, including 22 that are highly conserved. When mortality was assayed as a function of degree of fly-human conservation among all 134 tested genes (*Figure 2D*) we did observe greater mortality among highly conserved genes compared to less conserved genes. Not surprisingly, phylogenetic conservation indeed correlated positively with MI, a result that further supports using *Drosophila* as a model for functional evaluation of candidate genes for disease association.

In order to confirm that RNAi based phenotypes were indeed associated with target gene mRNA reduction, we measured RNA levels in heart tissue from flies expressing silencing constructs targeting six different genes from the top 26 group. These genes were chosen for analysis because they represent a range of different cellular functions, and include three genes for which silencing did not induce developmental lethality. MI values for the six genes ranged from normal to 43%. As shown in *Figure 3*, heart RNA levels of target genes were all significantly reduced. Furthermore, we tested developmental lethal phenotypes for all top 26 genes using two or more independent RNAi silencing fly lines for each gene, and obtained consistent MI results in all cases (data not shown).

Eight of the top 26 gene homologs encode products involved in histone H3K4 and H3K27 methylation (*Figures 2A* and *4*). Epigenetic activating (H3K4me) and inactivating (H3K27me) marks are associated with selective activation of promoters and enhancers, and chromatin marks are associated with specification and differentiation of cardiac cells (*Wamstad et al., 2012*), murine heart development (*He et al., 2012*; *Delgado-Olguín et al., 2012*), and cardiac pathogenesis (*Kaneda et al., 2009*; *Zhang and Liu, 2015*). Two additional top 26 genes (SUV420H1 and HUWE1) also encode conserved histone modifying proteins. Thus 42% of the top candidate genes from the PCGC study are involved in histone modification pathways, and all but one are highly conserved. Because chromatin modification pathway associated genes represent the major functional class among the 'top 26', we chose to examine in more detail the roles of these genes in *Drosophila* heart morphogenesis and function.

We analyzed heart phenotypes resulting from cardiac-specific silencing of the eight genes involved in H3K4 and H3K27 methylation (*Figure 4*). Silencing *kismet* yielded an adult heart phenotype in which the heart tube was severely affected (*Figure 4B*) with very low cardiac myofibrillar density (*Figure 4C*), elevated Pericardin (collagen) deposition (*Figure 4D*), and reduced numbers of cardioblasts (*Figure 4E*). Adult fly longevity was significantly affected, with approximately 50% of flies dead by day 25 (*Figure 4F*). During development, *kismet* silencing caused 54% mortality (*Figure 2A*). In larvae, cardiac myofibrillar density was reduced with pronounced heart tube filament disruption (*Figure 4G,H*), and Pericardin over-expression with disorganized ECM deposition (*Figure 4G,I*). *Trx*-silenced adult flies lacked cardiac actin and heart tube (*Figure 4B,C*), exhibited severely reduced Pericardin (*Figure 4D*), almost no cardioblast cells (*Figure 4E*), and all were dead at day 25 after emergence, compared to 20% mortality among control flies (*Figure 4F*). *Trx* silencing caused 35% developmental lethality (*Figure 2A*). In larvae, knock down of *trx* induced abnormal heart tube structure (*Figure 4G*) with reduced cardiac myofibrillar density (*Figure 4G,H*) and reduced Pericardin (*Figure 4G,I*). Silencing of *wds* caused complete developmental lethality (*Figure 2A*) and a dramatic larva heart phenotype (*Figure 4G*) featuring abnormal cardiac actin organization and fiber density (*Figure 4H*), and extremely high levels of Pericardin (*Figure 4G,I*). Taken together, these results strongly suggest that H3K4 methylation is critical for normal heart development and function, a conclusion supported by the observation that knockdown of the H3K4 de-methylase Lid had no phenotypic consequences (*Figure 4C–F,H,I*; *Figure 4—figure supplement 1*). Similarly, silencing of SMOX (involved in H3K27 de-methylation) resulted in minimal developmental mortality (MI 7%, *Figure 2A*) but normal cardiac myofibrillar density and Pericardin levels in larvae (*Figure 4H,I*). The adult heart phenotype was essentially normal, as was adult mortality at day 25 (*Figure 4*).

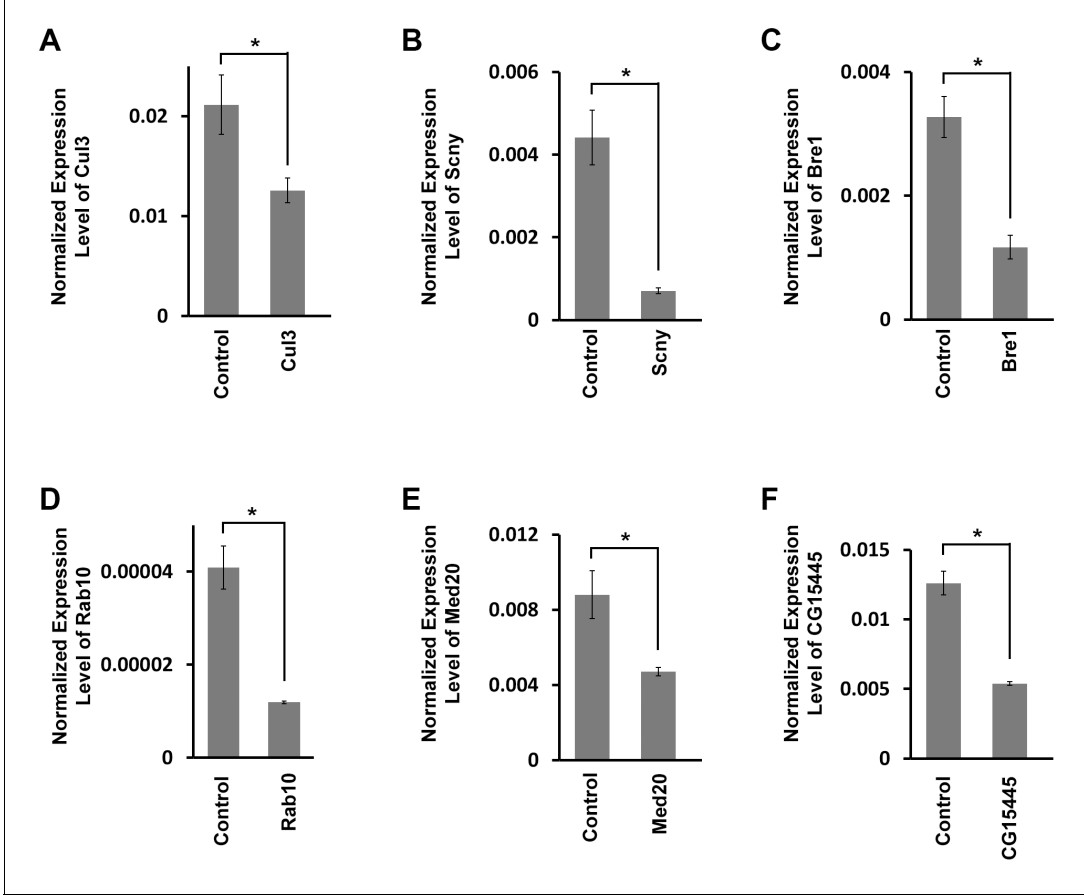

**Figure 3.** Target gene mRNA levels in adult fly heart tissues. (A) *Drosophila Cul3* homolog of human *CUL3* encoding ubiquitin ligase (Mortality Index 43%). (B) *Drosophila Scny* homolog of human *USP44* encoding ubiquitin protease (Mortality Index 25%). (C) *Drosophila Bre1* homolog of human *RNF20* encoding ubiquitin ligase E3 (Mortality Index 13%). (D) *Drosophila Rab10* homolog of human *RAB10* encoding GTPase (Mortality Index normal). (E) *Drosophila Med20* homolog of human *MED20* (Mortality Index normal). (F) *Drosophila CG15445* homolog of human *NUB1* (Mortality Index normal). mRNA levels in dissected adult hearts were determined by qRT-PCR. Statistical significance (*) was defined as p<0.05. Control flies were the progeny of a cross between homozygous 4XHand-Gal4 and *w1118*, which has one copy of 4xHand-Gal4 but does not carry a UAS-RNAi silencing construct.

Ubiquitination of H2BK120 (*Figure 4A*) is required for H3K4 methylation, and we observed that silencing of the *UbcD6* gene resulted in 84% developmental mortality (*Figure 2A*) and reduced adult longevity (*Figure 4F*; *Figure 4—figure supplement 1*). The adult heart was severely affected (*Figure 4—figure supplement 1*) with greatly reduced cardiac myofibrillar density (*Figure 4C*), elevated Pericardin (*Figure 4D*), and reduced cardioblast cell numbers (*Figure 4E*). Silencing of Bre1 and Scny also caused developmental mortality and cardiac myofibrillar density was somewhat reduced in larvae and adults (*Figure 4C,H*). Pericardin levels, cardioblast cell numbers, and adult mortality at day 25 were essentially normal (*Figure 4D,E,F,I*). By contrast, H3K4 de-methylation did not appear critical as evidenced by the essentially normal heart phenotype, normal MI, and normal adult longevity observed upon *Lid* silencing. (*Figure 4* and *Figure 4—figure supplement 1*). These results further demonstrate the utility of the *Drosophila* system to functionally test large-scale clinical genomics data and gain insights into fundamental pathways contributing to disease pathogenesis.

We used dye angiography to measure effects on heart function of cardiac-specific silencing of the eight genes involved in H3K4 and H3K27 methylation (*Figure 5*). Briefly, fluorescent dye was injected into the posterior abdomen of pharate (pre-eclosion) adult flies, where it entered the heart chamber and was pumped anteriorly. Dye accumulation in the head over a 30 s time interval provided a measurement of cardiac efficiency (*Figure 5A,B*) (*Drechsler et al., 2013*). We observed that in normal control flies fluorescent dye is readily detectable in the head within 15 s of injection, and

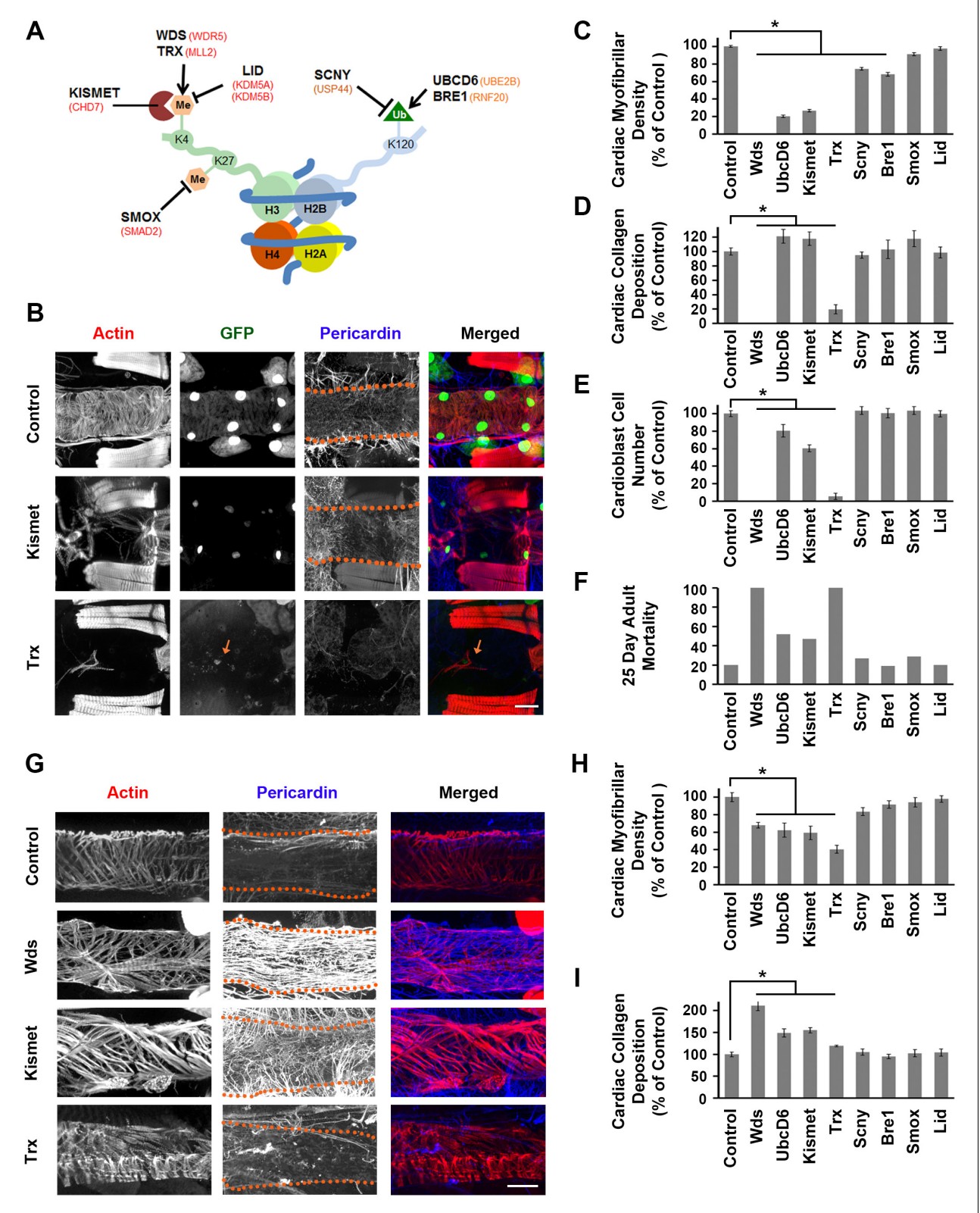

**Figure 4.** Genes involved in H3K4 and H3K27 methylation, mutated in CHD patients, affect heart structure, developmental mortality, and adult survival. (A) Depiction of nucleosome showing H3K4 and H3K27 methylation, and ubiquitination of H2BK120 (required for H3K4 methylation). *Drosophila* homologs involved in the production, removal, or interpretation of modifications are shown (human genes shown in red). (B) Adult heart phenotype induced by cardioblast-specific expression (driven by 4XHand-Gal4) of UAS-RNAi transgenes targeting *Kismet* and *Trx*. Cardiac actin (myofibers) was

*Figure 4 continued on next page*

*Figure 4 continued*

visualized by Phalloidin staining. Hand-GFP expression (nuclear) labels cardioblast cells. Pericardin was immune-labeled. Dotted lines delineate the heart tube outline. Red arrow points to remnant cardioblast cell in Trx-silenced heart. Scale bar = 50 μ. (C) Quantitation of adult heart cardiac myofibrillar density (as % of control; N = 10 for Control, N = 6 for indicated silenced gene). Statistical significance (*) was defined as $p < 0.05$. Scale bar = 50 μ. (D) Quantitation of adult heart cardiac collagen (Pericardin) deposition (as % of control; N = 10 for Control, N = 6 indicated silenced gene). Statistical significance (*) was defined as $p < 0.05$. (E) Quantitation of adult heart cardioblast cell numbers (cardioblasts expressing nuclear GFP; N = 10 for Control, N = 10 for indicated silenced gene). Statistical significance (*) was defined as $p < 0.05$. (F) Percentage of adult male flies dead at day 25 post-emergence (N = 50 flies per genotype). (G) Larva (third instar) heart phenotypes induced by cardioblast-specific expression (driven by 4XHand-Gal4) of UAS-RNAi transgenes targeting *Wds*, *Kismet*, and *Trx*. Cardiac actin (myofibers) was visualized by Phalloidin staining. The pericardin was immune-labeled. Dotted lines delineate heart tube outline. (H) Quantitation of larval heart cardiac myofibrillar density (as % of control; N = 10 for Control, N = 6 for indicated silenced gene). Statistical significance (*) was defined as $p < 0.05$. (I) Quantitation of larva heart cardiac collagen (Pericardin) deposition (as % of control; N = 10 for Control, N = 6 for indicated silenced gene) Statistical significance (*) was defined as $p < 0.05$. Control flies were the progeny of a cross between homozygous 4XHand-Gal4 and $w^{1118}$, which has one copy of 4xHand-Gal4 but does not carry a UAS-RNAi silencing construct.

The following figure supplement is available for figure 4:

**Figure supplement 1.** Genes involved in H3K4 and H3K27 methylation, mutated in CHD patients.

significantly accumulated by 30 s. By contrast, silencing of candidate CHD gene homologs involved in the methylation of H3K4 and H3K27 profoundly impaired cardiac function (*Figure 5B,C*). Silencing of *Smox* and *Lid*, however, did not measurably impair heart function as measured in this assay. These observations indicate that H3K4 and H3K27 methylation, and not demethylation, is essential for heart function, consistent with our findings with regard to cardiac tissue morphology and effects on developmental lethality and adult longevity.

Gene silencing using available and well-established UAS-RNAi fly lines allows rapid and efficient functional screening of large numbers of candidate genes. We also established an in vivo platform to analyze the cardiac phenotypes of specific patient-derived alleles of candidate CHD genes in *Drosophila*, employing a 'gene replacement' strategy. As a proof-of-concept, we tested the ability of a wild type human WDR5 allele to rescue heart phenotypes induced by silencing of the endogenous *wds Drosophila* homolog. Heart-specific silencing of *wds* caused 100% developmental lethality and abnormal heart morphology in late larvae characterized by reduced cardiac myofibers and severe over-deposition of Pericardin (*Figure 6*). We found that simultaneously overexpressing a wild type human WDR5 allele reduced developmental lethality almost 7-fold, significantly restored cardiac myofibrillar density, and reduced Pericardin levels essentially to normal (*Figure 6*). By contrast, when the endogenous *wds* expression was 'replaced' by a CHD patient-derived WDR5-K7Q mutant allele, developmental lethality remained quite elevated, cardiac myofibrillar density remained abnormally low (relative both to Control and wild type WDR5 rescue), and Pericardin levels were lowered but still significantly higher than both Control and wild type WDR5 rescue. These observations illustrate the structure-function homologies between human and fly 'heart genes' that support use of *Drosophila* for functional validation of candidate CHD genes. Moreover, they demonstrate that the 'gene replacement' strategy can be used to quantitatively assess phenotypes induced by patient-specific alleles of candidate disease genes.

## Discussion

Our results demonstrate the value of the *Drosophila* model system to functionally screen and validate candidate disease genes identified through large scale sequencing efforts. The high-throughput testing platform described here can extend gene association data to quantitative, in vivo morphological and functional criteria. On the basis of this screening, the most promising genes can justifiably be advanced to more costly, time consuming, and technically challenging validation platforms required to definitively identify and confirm disease-causing gene variants. Such advanced testing, we presume, will encompass gene function analysis in a mammalian model system.

The advantages of functional screening using RNAi-based silencing lines are to some extent offset by certain limitations. From the outset, of human disease genes, approximately 25% are not represented by *Drosophila* homologs. False negatives may occur if RNAi expression does not

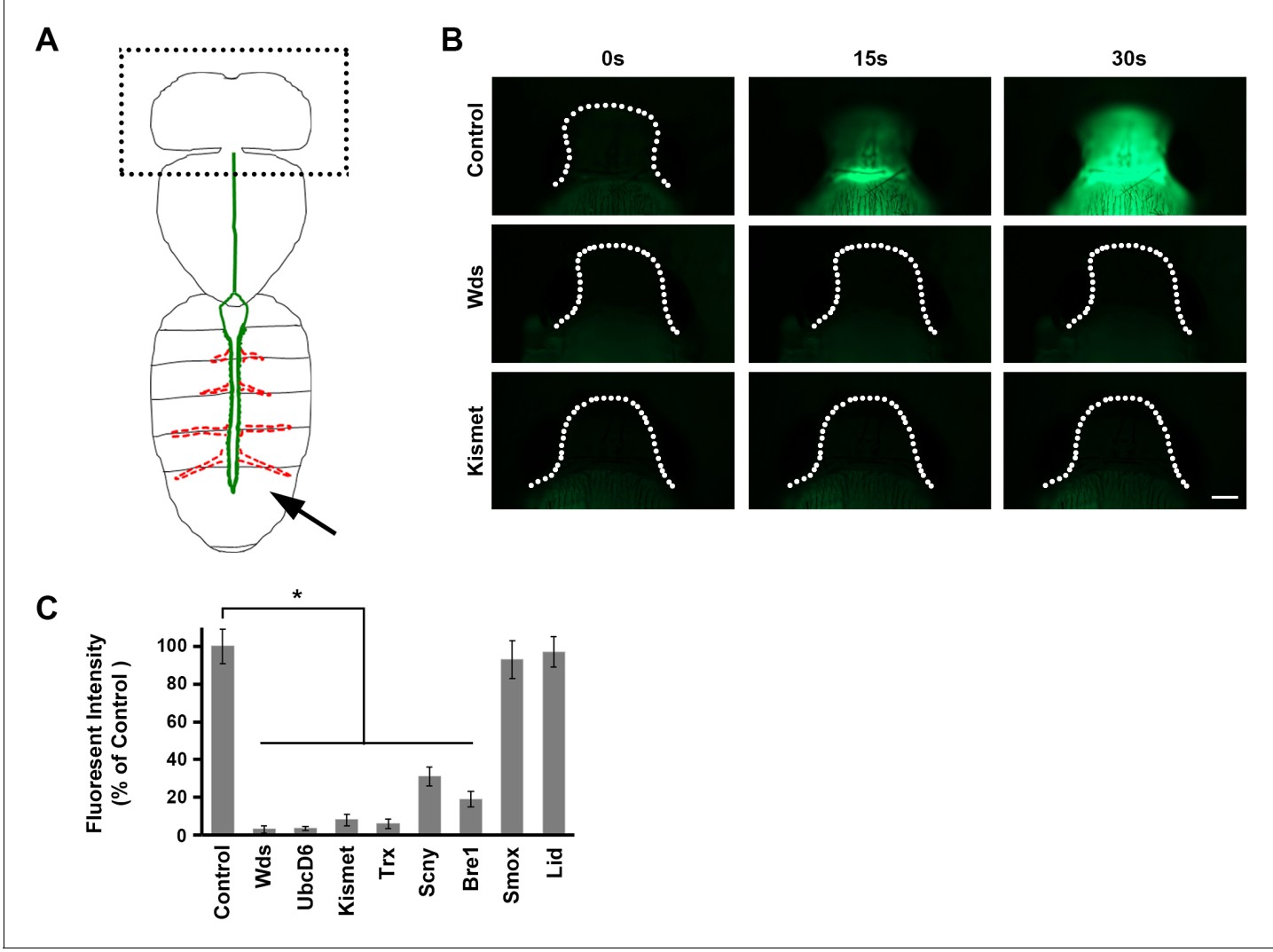

**Figure 5.** Direct measurement of heart function. (A) The adult heart is depicted schematically in green. Supporting lateral alary muscles are depicted in red. Fluorescent dye injected into the posterior body cavity (arrow) enters the heart tube lumen and is pumped anteriorly into the brain region. (B) Time course of injected fluorescent dye entering the brain region (dotted outline). In normal control flies dye is easily detectable 15 s (s) after injection, and the brain is highly fluorescent within 30 s. In flies expressing heart-specific *Wds* or *Kismet* gene targeting RNAi silencing constructs, by contrast, dye does not reach the brain by 30 s after injection. Scale bar = 100 μ. (C) Quantitative analysis of brain region fluorescence, relative to control fly levels, as a function of heart-specific gene silencing. Experiments were performed in triplicate (3 independent dye injection experiments). Statistical significance (*) was defined as p<0.05. Control flies were the progeny of a cross between homozygous 4XHand-Gal4 and *w^1118^*, which has one copy of 4xHand-Gal4 but does not carry a UAS-RNAi silencing construct.

sufficiently reduce mRNA levels. However, we tested a number of RNAi lines by qRT-PCR and found no examples of ineffective gene silencing. False negatives may also occur when missense mutations produce gain-of-function (GOF) alleles, and we cannot accurately predict the degree to which this could confound our analysis. This problem, we note, is mitigated to a degree by the fact that genes altered by GOF mutations would in many instances *normally* encode proteins with cardiac cell functions, and thus yield silencing induced phenotypes in our functional screening system. False positives can arise from so-called 'off target' effects of a given RNAi targeting construct. We controlled against misleading results from both RNAi silencing inefficiency and off target effects by employing the latest-generation available RNAi lines, and testing multiple such lines for each of the top 26 candidate genes. This precaution against false negative and false positive results further limited the total number of candidate genes we could screen to 134 (60% of the CHD candidate genes published by

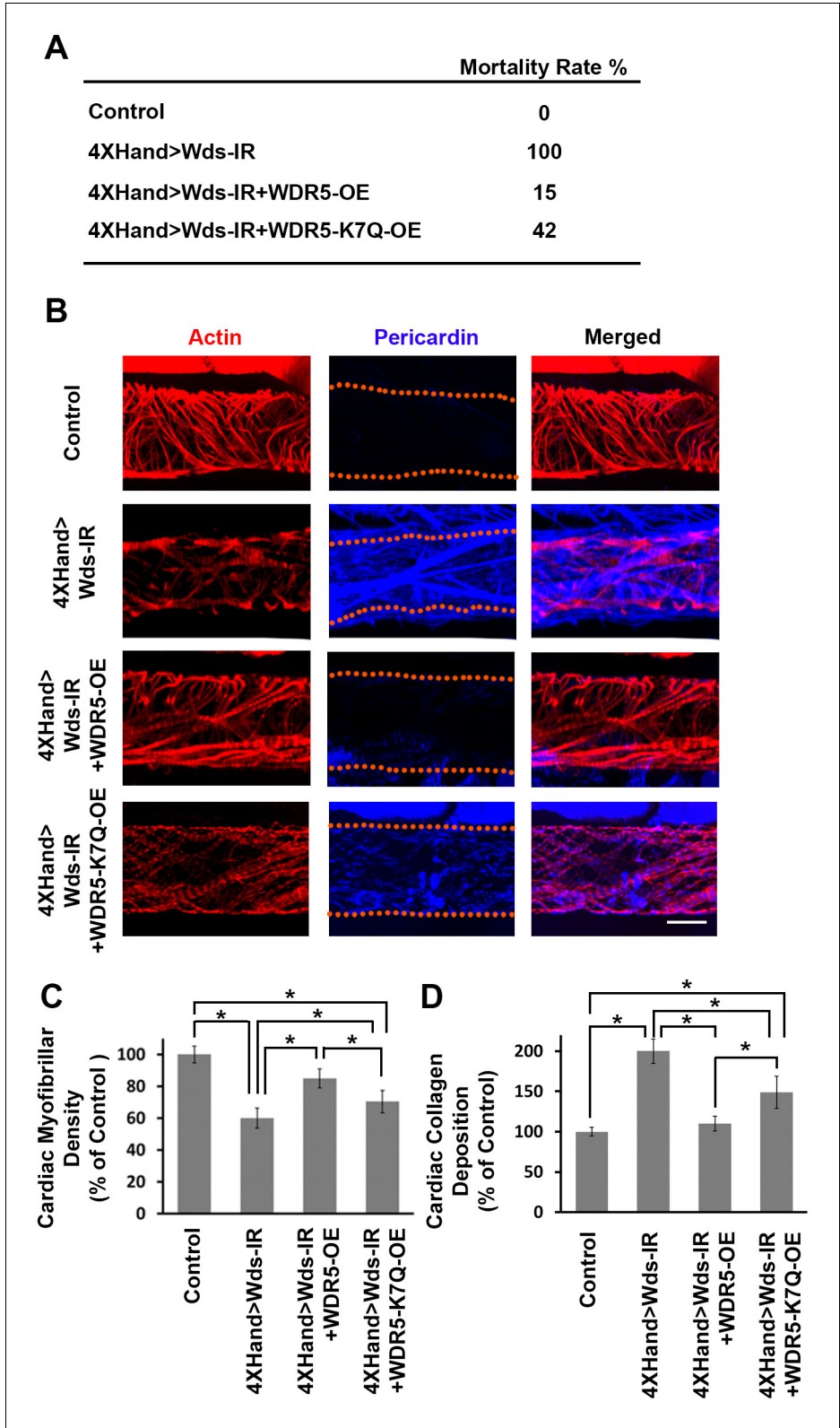

**Figure 6.** *Wds* silencing-induced lethality and heart phenotypes rescued by wild type human *WDR5* but not by a patient derived *WDR5-K7Q* mutant allele. (**A**) Developmental lethality (Mortality Index) for flies in which endogenous *Wds* heart expression was silenced, and attempted rescue by either wild type *WDR5* or mutant *WDR5-K7Q* overexpression (OE). (**B**) Larva (third instar) heart phenotype induced by cardioblast-specific expression

*Figure 6 continued on next page*

*Figure 6 continued*

(driven by 4XHand-Gal4) of UAS-RNAi transgene targeting *Wds*, and attempted rescue by *WDR5* or *WDR5-K7Q* overexpression. Cardiac actin (myofibers) visualized by Phalloidin staining. Pericardin was immune-labeled. Dotted lines delineate the heart tube outline. (C) Quantitation of larval heart cardiac myofibrillar density (as % of control; N = 10 for Control, N = 6 for Wds-IR, Wds-IR+WDR5-OE, or Wds-IR+WDR5-K7Q-OE). Statistical significance (*) was defined as p<0.05. For each strain, fiber density was significantly lower than control. Rescue by wild type WDR5 significantly rescued fiber density, to levels significantly greater than achieved by the overexpression of the WDR5-K7Q mutant allele. (D) Quantitation of larva heart cardiac collagen (Pericardin) deposition (as % of control; N = 10 for Control, N = 6 for Wds-IR, Wds-IR+WDR5-OE, or Wds-IR+WDR5-K7Q-OE) Statistical significance (*) was defined as p<0.05. Wild type WDR5-OE fully rescued up-regulated collagen deposition induced by Wds-IR. Mutant WDR5-K7Q-OE partially rescued the Wds-IR phenotype, but collagen deposition levels remained upregulated compared to the control. The control flies were the progeny of a cross between homozygous 4XHand-Gal4 and *w^1118*, which has one copy of 4xHand-Gal4 but does not carry a UAS-RNAi silencing construct.

the PCGC). Nevertheless, our in vivo screening system rapidly and efficiently confirmed roles in heart development, tissue maintenance, or cardiac function for 72 candidate genes.

Phenotypic analysis based on gene silencing, while effective for high throughput screening, represents a first step in validation and characterization of candidate mutations. We also demonstrated an initial 'gene replacement' strategy that takes further advantage of the available resources and sophisticated genetics of the *Drosophila* model system to quantitatively characterize the phenotype induced by expression of a specific patient-derived mutant allele. This approach may prove highly valuable as a testing platform for precision medicine based therapeutic drugs. Ultimately, the knowledge that a specific mutation contributes to disease pathogenesis opens the door to the application of precise gene-editing interventions, strategies for which are currently being developed based on CRISPR/Cas-9 mediated genome editing (*Long et al., 2016*; *Carroll et al., 2016*; *Long et al., 2014*). The high throughput candidate gene validation studies reported here represent the essential first step in the functional confirmation, phenotypic characterization, and ultimate precision treatment of disease-associated mutations.

## Materials and methods

### Fly stocks
*Drosophila* lines were obtained from the Bloomington Stock Center (Bloomington, IN; NIH P40OD018537). Transgenes were overexpressed with the *UAS-GAL4* system (*Brand and Perrimon, 1993*). To increase the penetrance of heart-specific UAS-RNAi mediated gene silencing we developed a powerful Gal4 driver incorporating four repeats of the *Hand* gene (*Han et al., 2006*) enhancer. 4XHand-Gal4 was generated by tandem insertions of the original Hand cardiac-specific enhancer (*Han and Olson, 2005*) in the pGal4 vector.

### DNA cloning and generation of transgenic fly strains
The wild type *WDR5* cDNA was obtained from OriGene, which encodes the 334 a.a. protein with GenBank ID P61964. The *WDR5-K7Q* cDNA was generated by introducing the K7Q mutation into the wild type *WDR5* cDNA sequence using PCR. To generate *UAS-WDR5* and *UAS-WDR5-K7Q* constructs, the above cDNAs were cloned into the *pUASTattB* vector and the transgenes were introduced into a fixed chromosomal docking site by germ line transformation.

### Dye angiography
Pharate (pre-eclosion) adult flies were carefully removed from pupa cases and affixed by double-sided scotch tape to glass slides. A single injection of 100 nl of uranin solution (1 µg/µl in PBS) was delivered into the abdomen using a Drummond Nanoject II auto-nanoliter injector with a glass capillary. Dye accumulation in the fly head was monitored over a 3 min interval using a Zeiss ApoTome.2 microscope with a 5x Plan-Neofluar 0.16 air objective. Experiments were performed in triplicate (3 independent dye injection experiments). ImageJ software Version 1.49 was used for image processing and quantification. Fluorescence intensity in the fly head was measured at times 0 (time of

injection) and 30 s post-injection. Increased fluorescence values represent fluorescence at t = 30 s minus t = 0, and were expressed relative to head fluorescence of control flies not expressing an RNAi targeting transgene.

### qRT-PCR analysis

RNA was isolated using Trizol Reagent (Invitrogen, Carlsbad, CA) from heart tissue dissected from 60 adult flies of the relevant genotype. RNA purity and concentration were determined using a Nanodrop-1000 (Thermo Scientific, Wilmington, DE). Total RNA (1 µg) was reverse transcribed using Superscript IV (Invitrogen). SYBR Green based real-time qPCR (Power Cyber Mastermix; Applied Biosystems, Carlsbad, CA) was performed using a StepOne Plus (Applied Biosystems). Gene-specific primer pairs were used. Quantitative values were determined using the $2^{-\Delta\Delta CT}$ method (*Livak and Schmittgen, 2001*), normalizing to Gapdh. Values are derived from three technical replicates of qRT-PCR experiments from pooled RNA.

### Fly imaging

Fly preparation and imaging are described in more detail at Bio-protocol (*Zhu et al., 2017*). Larvae and adult flies were dissected and fixed for 10 min in 4% paraformaldehyde in phosphate-buffered saline (PBS). Alexa Fluor 555 Phalloidin was obtained from Thermo Fisher. Mouse anti-Pericardin antibody (EC11) was used at 1:500 dilutions, followed by Cy3-conjugated secondary antibodies (Jackson Lab). Confocal imaging was performed with a Zeiss ApoTome.2 microscope using a 20× Plan-Apochromat 0.8 N.A. air objective. For quantitative comparisons of intensities, common settings were chosen to avoid oversaturation. ImageJ Software Version 1.49 was used for image processing. For quantitative comparisons of cardiac myofibrillar density, cardioblast cell numbers, and Pericardin deposition we analyzed six to ten control flies and six to ten flies of each experimental genotype. These features were quantitatively assessed in the same area of the same segment in each fly. Cardiac myofiber levels of Z-stack projections were carefully selected for analysis, avoiding the ventral muscle layer underlying the heart tube. Sample size determinations were based upon extensive previous experience in analyzing fly heart morphology.

### Survival assay

Within 24 hr of egg laying *Drosophila* larvae were transferred from 25°C to 29°C to boost UAS-transgene expression. Adult male flies were maintained at 29°C in groups of 15 or fewer; 50 flies in total were assayed per genotype.

### Statistical analysis

Statistical tests were performed using PAST.exe software (http://folk.uio.no/ohammer/past/index.html) unless otherwise noted. Sample errors are given as standard error of the mean (s.e.m). Data were first tested for normality by using the Shapiro-Wilk test ($\alpha = 0.05$). Normally distributed data were analyzed either by Student's *t*-test (two groups) and Bonferroni comparison to adjust p value or by a one-way analysis of variance followed by a Tukey-Kramer post-test for comparing multiple groups. Non-normal distributed data were analyzed by either a Mann-Whitney test (two groups) and Bonferroni comparison to adjust the p value or a Kruskal-Wallis H-test followed by a Dunn's test for comparisons between multiple groups. Statistical significance was defined as $p < 0.05$.

## Acknowledgements

We thank the Bloomington Drosophila Stock Center at Indiana University and the Vienna Drosophila Resource Center for providing the fly stocks. We thank the Developmental Studies Hybridoma Bank at the University of Iowa for antibodies. Z.H. was supported by grants from the National Institutes of Health (RO1-HL090801, RO1-NK098410).

## Additional information

### Funding

| Funder | Grant reference number | Author |
|---|---|---|
| National Heart, Lung, and Blood Institute | NIH R01 | Zhe Han |
| National Institute of Diabetes and Digestive and Kidney Diseases | NIH R01 | Zhe Han |

The funders had no role in study design, data collection and interpretation, or the decision to submit the work for publication.

### Author contributions

J-yZ, YF, Data curation, Formal analysis, Validation, Investigation, Visualization, Methodology, Writing—original draft; MN, Data curation, Formal analysis, Investigation, Methodology; AR, Data curation, Validation, Investigation, Methodology, Writing—original draft; ZH, Conceptualization, Resources, Data curation, Formal analysis, Supervision, Funding acquisition, Validation, Investigation, Methodology, Writing—original draft, Project administration, Writing—review and editing

### Author ORCIDs

Zhe Han, http://orcid.org/0000-0002-5177-7798

## Additional files

### Supplementary files

• Supplementary file 1. 134 de novo mutated genes from CHD study participants with corresponding *Drosophila* homologs showing conservation score (*Hu et al., 2011*), function (from Flybase) and mortality index (determined by crossing homozygous UAS-RNAi transgenic flies with flies bearing a 4XHand-Gal4 driver balanced over *CyO*, progeny flies that emerge as adults with curly wings (*CyO*, no transgene expression) vs. straight wings (4XHand-Gal4 driven UAS-RNAi transgene expression in cardioblasts) are recorded and the lethal rate attributable to RNAi heart expression is calculated as (Curly – Straight / Curly) X 100 = % Mortality). Patient identification number, human gene, mutation type and category (DH, *d*amaging mutation in gene with homolog expression *h*igh in embryonic day 14.5 mouse heart (emh); CH, mutation in phylogenetically *c*onserved region of gene with homolog expression *h*igh in emh; NH, mutation in phylogenetically *n*onconserved region of gene with homolog expression *h*igh in emh; DL, *d*amaging mutation in gene with homolog expression *l*ow in emh; CL, mutation in phylogenetically *c*onserved region of gene with homolog expression *l*ow in emh; NL, mutation in phylogenetically *n*onconserved region of gene with homolog expression *l*ow in emh) are from (*Zaidi et al., 2013*).

• Supplementary file 2. *Drosophila* lines used to silence expression of genes. Flies were obtained from either the Bloomington *Drosophila* Stock Center (BDSC) or the Vienna *Drosophila* Resource Center (VDRC). Individual fly lines are identified by source ID numbers. Both the human genes and their corresponding *Drosophila* homologs are listed. Each line carries an inhibitory RNA transgene designed to silence the indicated *Drosophila* gene, cloned downstream of a UAS DNA element. The RNAi transgene is only expressed in cells producing the yeast Gal4 transcription factor.

• Supplementary file 3. Gene specific PCR primers.

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
