## [Decision Letter]

Thank you for submitting your article "High throughput in vivo functional validation of candidate congenital heart disease genes in *Drosophila*" for consideration by *eLife*. Your article has been reviewed by three peer reviewers, one of whom is a member of our Board of Reviewing Editors, and the evaluation has been overseen by K VijayRaghavan as the Senior Editor. The reviewers have opted to remain anonymous.

The reviewers have discussed the reviews with one another and the Reviewing Editor has drafted this decision to help you prepare a revised submission.

Summary:

This is a significant paper which demonstrates the effectiveness of using *Drosophila* as a rapid screening tool for assessing the possible functional significance of genes identified in genomic screens, e.g. screens for de novo variants associated with congenital heart disease as in Zaidi et al. 2013. Genome-wide RNAi screens have been used previously to identify pathways important for cardiovascular development and cardiac disease in humans (e.g. Kim et al., 2010; Neely et al., 2010; van der Harst et al. 2016), so the approach itself is not particularly novel. In the design used here two RNAis for candidate CHD genes are used to knock down the *Dros* homologues of disease genes in Hand-positive cardioblasts using for the first time a multimerized (4x) Hand enhancer driving Gal4, a design that likely leads to high efficiency knockdown compared to previous approaches. A number of cardiac-related phenotypes were scored, including mortality, adult longevity, muscle fibre density, cardioblast numbers, collagen deposition, and heart tube function. The authors perform screens of the top 26 genes identified by Zaidi et al. including the genes involved in chromatin modification, as well as the larger set of candidate genes for which there were *Dros* homologues. They found a high degree of mortality associated with both sets. Interestingly, knockdown of the top 26 did not lead to more severe mortality than the larger set, potentially highlighting the limitations of bioinformatics alone in identifying the most significant players in human disease and the need to combine with functional analysis. Knockdown of a number of chromatin modifying genes were characterised in more detail. The authors conclude from these results that histone methylation but not demethylation, is essential for heart development in flies. Rescue of the *Dros* knockdown of WDR5 with the human WT gene but not the same gene carrying the candidate disease variant was also presented in a proof-of-principal study.

The work is timely and elegant. Clearly knocking down of genes in fly development as a surrogate for testing the relevance of single amino acid variants present in patients with human disease has its limitations and some of these are discussed, but this is clearly a powerful system for focusing attention on variants with significant affects in this assay for follow up functional studies in mammalian cells or animal models.

A number of issues were identified by the reviewers and those requiring attention are listed below:

Essential revisions:

1) Could over-expression of 4xHand-Gal4 sensitize flies to RNAi knockdown phenotypes? How would this be tested?

2) We disagree that gain of function variants will be rare. New insights into this for transcription factors has come to light – see for example Luna-Zurita et al. Cell 2016 and Bouveret et al. *eLife* 2015 – even apparently null proteins can have gain of function effects. There are in fact a number of ways through which interruption of conserved domains in multidomain proteins may have gain of function effects. I think this should be considered more carefully in the Discussion in the context of limitations of the approach.

3) One issue that should be addressed is the rationale for the choice of genes for the various analyses. Figure 3 tests the expression levels of six genes when knocked down, but the choice of these genes, and why all 26 were not tested, is not clear. In fact there are certain biases in this study in the sense that one might expect severe defects in any cell type expressing loss or gain of function alleles for chromatin modifying genes, given their global function in epigenetic regulation. We accept that these examples serve to highlight the phenotyping pathway used and the range of defects that can be generated. However, the paper would be strengthened by including some examples of candidates for the Zaidi study where there is mortality or a strong heart phenotype when the gene is knocked down that reveals its totally unknown functions in heart development and cellular function.

4) Overall, the description of Methods is seriously lacking in detail. The authors should indicate all of the primer sequences that were used, and all of the RNAi lines should be listed in order that the results can be replicated accurately. The dye angiography method does not include detail about how the data was analyzed.

5) We are not convinced that appropriate statistical tests were used when comparing multiple genotypes. Our understanding is that simple t-tests might not be appropriate, but analysis of variance should initially be used. Moreover, simple t-tests can only be used when the sample variances are comparable. An expert statistical review is advised.

6) It is mentioned that "Interestingly, the "top 26" genes identified by the PCGC study all have conserved *Drosophila* homologs, but the involvement of these genes in *Drosophila* heart development had not previously been assessed." This is inaccurate. Neely et al. (2010) investigated 18/26 of these genes, many of which when silenced using a similar cardiac-specific approach, reduced lifespan. While all were screened for developmental lethality, only one was categorized as a "developmentally lethal gene". The low hit number was potentially due to inadequate RNAi-mediated knockdown relative to that afforded by the 4X-Hand driver described here. Nonetheless, involvement of ~70% of the 26 genes in cardiac development was indeed assessed, and the fact that this highly-related manuscript was not cited and the overlap clarified needs correction.

7) For all analysis depicted in Figure 4–Figure 6, information related to the control line(s) or the RNAi stock numbers used, is not provided. This is critical for data interpretation and to rule out genetic background effects, which can contribute to differences in structure/physiology. If the experiments are inappropriately controlled, they must be repeated accordingly. For example, 1118 is not an ideal control for the experiments outlined in Figure 3.

8) Unlike developmental/survival analysis following cardiac-restricted RNAi expression, structural assessments, and fluorescent dye pumping capacity, which have all been performed previously by others, the gene replacement strategy, involving heart-specific silencing of fly genes and simultaneous expression of normal or mutant human alleles, is exceptionally intriguing. Unfortunately, excitement is severely tempered due to the use of standard P element-mediated germline transformation. First, we must assume the RNAi does not target the products of the human alleles, since no information regarding design is presented. This should be demonstrated. Second, no data related to transgene expression is given. Perhaps the WDR5-K7Q has lower expression levels than WDR5 and, hence, a distinct rescue potential? Finally, what about insertional effects? Were results consistent among multiple transgenic lines? Use of site-specific recombinase technologies and verification of equivalent transgene expression levels are recommended.

[Editors' note: further revisions were requested prior to acceptance, as described below.]

Thank you for resubmitting your work entitled "High throughput in vivo functional validation of candidate congenital heart disease genes in *Drosophila*" for further consideration at *eLife*. Your revised article has been evaluated and discussed by K VijayRaghavan as the Senior editor and a Reviewing editor.

The manuscript has been improved but there are some remaining issues that need to be addressed before acceptance, as outlined below:

The editors remain concerned about the responses to points 7 and 8 in your itemised responses to reviewer's concerns. Specifically, they note a completely different narrative has been submitted in response to concerns about 1. the nature of the control lines uses for the study; and 2. the uses of P-element versus targeted chromosome integration for generation of transgenic lines. The revisions put forward are completely different from what was originally submitted. At a minimum, these "errors" appear to reflect a sloppiness in the preparation of the manuscript that remains of concern to the editors. For this manuscript to be considered further, the authors must provide a reason and concrete evidence that these major technological issues were mistakenly recorded incorrectly.

---

## [Author Response]

*Essential revisions:*

*1) Could over-expression of 4xHand-Gal4 sensitize flies to RNAi knockdown phenotypes? How would this be tested?*

This is formally possible but we do not perceive an obvious mechanism for how or why this might occur, as we understand the question, across the fly genome. As a result we did not directly test this possibility. However, the fact that our unbiased screen (using 4xHand-Gal4) of approximately 800 *Drosophila* genes yielded only 6% developmental lethals strongly suggests that 4xHand-Gal4 expression did not sensitize flies to knockdown phenotypes. Text addressing this point has now been added to Results section.

*2) We disagree that gain of function variants will be rare. New insights into this for transcription factors has come to light – see for example Luna-Zurita et al. Cell 2016 and Bouveret et al. eLife 2015 – even apparently null proteins can have gain of function effects. There are in fact a number of ways through which interruption of conserved domains in multidomain proteins may have gain of function effects. I think this should be considered more carefully in the Discussion in the context of limitations of the approach.*

We are grateful to the reviewers for raising this point and directing our attention to these insightful articles. In response, we have addressed this point more carefully in the Discussion in the context of limitations of our approach.

*3) One issue that should be addressed is the rationale for the choice of genes for the various analyses. Figure 3 tests the expression levels of six genes when knocked down, but the choice of these genes, and why all 26 were not tested, is not clear. In fact there are certain biases in this study in the sense that one might expect severe defects in any cell type expressing loss or gain of function alleles for chromatin modifying genes, given their global function in epigenetic regulation. We accept that these examples serve to highlight the phenotyping pathway used and the range of defects that can be generated. However, the paper would be strengthened by including some examples of candidates for the Zaidi study where there is mortality or a strong heart phenotype when the gene is knocked down that reveals its totally unknown functions in heart development and cellular function.*

We have now added text to the Results explaining our selection of the six genes tested in the experiments illustrated in Figure 3. We did not test all 26 genes by quantitative PCR because we tested multiple RNAi lines for all of them, and in each case observed consistent phenotype outcomes. We have also added text to emphasize why we chose to focus more detailed analysis on the phenotypic consequences of knocking down genes involved in chromatin modification pathways. We agree with the reviewer that one might expect to see severe defects from loss of function of chromatin modifying genes in any cell type, but in fact we observed a normal heart phenotype upon silencing of Lid, one of the reasons for examining this gene by qRT-PCR analysis. To address the reviewer’s comment that the paper would be strengthened by revealing heretofore unknown functions of candidate genes, we respectfully submit that ours is a functional validation study, rather than a novel gene identification study (as clearly indicated by the title, "High throughput in vivo functional validation of candidate congenital heart disease genes in *Drosophila*"). Therefore, we argue that in this sense the reviewer’s suggestion is outside the scope of our current submission.

*4) Overall, the description of Methods is seriously lacking in detail. The authors should indicate all of the primer sequences that were used, and all of the RNAi lines should be listed in order that the results can be replicated accurately. The dye angiography method does not include detail about how the data was analyzed.*

We now provide information on all primer sequences ([Supplementary-material SD3-data]) and list all RNAi lines ([Supplementary-material SD2-data]). We have added more details and information on how the dye angiography data was analyzed.

*5) We are not convinced that appropriate statistical tests were used when comparing multiple genotypes. Our understanding is that simple t-tests might not be appropriate, but analysis of variance should initially be used. Moreover, simple t-tests can only be used when the sample variances are comparable. An expert statistical review is advised.*

We have now added a “Statistical analysis” section to the Materials and methods, providing much more detail on the statistical tests we used to analyze our data. We have consulted with a statistics expert to review our methods.

*6) It is mentioned that "Interestingly, the "top 26" genes identified by the PCGC study all have conserved Drosophila homologs, but the involvement of these genes in Drosophila heart development had not previously been assessed." This is inaccurate. Neely et al. (2010) investigated 18/26 of these genes, many of which when silenced using a similar cardiac-specific approach, reduced lifespan. While all were screened for developmental lethality, only one was categorized as a "developmentally lethal gene". The low hit number was potentially due to inadequate RNAi-mediated knockdown relative to that afforded by the 4X-Hand driver described here. Nonetheless, involvement of ~70% of the 26 genes in cardiac development was indeed assessed, and the fact that this highly-related manuscript was not cited and the overlap clarified needs correction.*

We are very grateful to the reviewers for pointing this out. We have added text addressing the overlap, and appropriately cite the related publication.

*7) For all analysis depicted in Figure 4–Figure 6, information related to the control line(s) or the RNAi stock numbers used, is not provided. This is critical for data interpretation and to rule out genetic background effects, which can contribute to differences in structure/physiology. If the experiments are inappropriately controlled, they must be repeated accordingly. For example, w^1118^ is not an ideal control for the experiments outlined in Figure 3.*

We have added text to clarify the nature of the control lines and provide a comprehensive inventory of all RNAi lines used ([Supplementary-material SD2-data]). We also made changes to the labels in all figures that used control flies, and added to the figure legends to clearly state the genotype of each groups of control flies.

*8) Unlike developmental/survival analysis following cardiac-restricted RNAi expression, structural assessments, and fluorescent dye pumping capacity, which have all been performed previously by others, the gene replacement strategy, involving heart-specific silencing of fly genes and simultaneous expression of normal or mutant human alleles, is exceptionally intriguing. Unfortunately, excitement is severely tempered due to the use of standard P element-mediated germline transformation. First, we must assume the RNAi does not target the products of the human alleles, since no information regarding design is presented. This should be demonstrated. Second, no data related to transgene expression is given. Perhaps the WDR5-K7Q has lower expression levels than WDR5 and, hence, a distinct rescue potential? Finally, what about insertional effects? Were results consistent among multiple transgenic lines? Use of site-specific recombinase technologies and verification of equivalent transgene expression levels are recommended.*

We apologize profusely for mistakenly describing our transgenic lines as having been produced by standard P element-mediated transformation. This was incorrect! In fact, our gene replacement strategy involving heart-specific silencing of fly genes and simultaneous expression of normal or mutant human alleles used a *pUASTattB* vector for fixed chromosomal docking site to effect germline transformation. We have corrected the text of Materials and methods accordingly and hope that in so doing we have restored reviewer excitement. The RNAi target sequence silencing the endogenous *Drosophila* WDS gene is not expected to silence WDR5, since there is over one third of the targeting sequence are mismatch when compared to the human WDR5 sequence. In addition, phenotypic rescue effects from human cDNA expression indicated that even if silencing of human genes did occur at very low level it did not prevent our analysis, since it will have same effects to both the wild type and the mutant allele of human WDR5. Insertional effects is no longer an issue since both the wild type and the mutant allele of human WDR5 were indeed inserted to the same chromosome location and should have the same level of expression.

[Editors' note: further revisions were requested prior to acceptance, as described below.]

*Thank you for resubmitting your work entitled "High throughput* in vivo functional validation of candidate congenital heart disease genes in Drosophila" for further consideration at eLife. Your revised article has been evaluated and discussed by K VijayRaghavan (Senior editor) and the Reviewing editor.

*The manuscript has been improved but there are some remaining issues that need to be addressed before acceptance, as outlined below:*

*The editors remain concerned about the responses to points 7 and 8 in your itemised responses to reviewer's concerns. Specifically, they note a completely different narrative has been submitted in response to concerns about 1. the nature of the control lines uses for the study; and 2. the uses of P-element versus targeted chromosome integration for generation of transgenic lines. The revisions put forward are completely different from what was originally submitted. At a minimum, these "errors" appear to reflect a sloppiness in the preparation of the manuscript that remains of concern to the editors. For this manuscript to be considered further, the authors must provide a reason and concrete evidence that these major technological issues were mistakenly recorded incorrectly.*

We are deeply appreciative and grateful for the continued consideration of our manuscript “High throughput in vivo functional validation of candidate congenital heart disease genes in *Drosophila*”, and the concern for credible evidence and reasoning to show that in addressing points 7 and 8 of the itemized reviewer concerns we indeed addressed erroneously recorded information and that the text now accurately describes the actual techniques and approaches that were employed. We do sincerely apologize for these oversights that occurred in the initial preparation of our manuscript, and agree that addressing such sloppiness indeed requires extended care and consideration on our part to overcome the completely justified concerns of the editors.

To clarify what occurred with respect to describing the “nature of the control lines uses for the study”: *w^1118^* was used throughout the course of our experiments as an abbreviated form of “4xHand-Gal4 crossed to *w^1118^*” which always served as our control. This practice was internally consistent with our use of “GeneX” as an abbreviated form of “4xHand-Gal4 crossed to UAS-GeneX-RNAi” as a descriptor of the experimental test lines that were being analyzed in large numbers during the course of our study. Unfortunately, use of the *w^1118^*abbreviation for the control lines was used in our initial manuscript preparation, and proved misleading. We are grateful for the editor’s detection of this oversight, and for this second opportunity to clarify the nature of the control lines and guarantee the accurate reporting and interpretation of our findings.

With respect to concern regarding “the uses of P-element versus targeted chromosome integration for generation of transgenic lines”: indeed “targeted chromosome integration” is the superior approach in this case to use of “P-element mediated random integration” and this experiment was in fact carried out using a targeted chromosome integration approach. We apologize that in initially preparing our manuscript we erroneously inserted previously drafted text from a different study’s “Materials and methods” section, then failed to make the changes required to correctly describe our approach. Again, we sincerely apologize for this sloppiness.